# Effects of Low Dietary Cation and Anion Difference on Blood Gas, Renal Electrolyte, and Acid Excretions in Goats in Tropical Conditions

**DOI:** 10.3390/ani12233444

**Published:** 2022-12-06

**Authors:** Dang Khoa Do Nguyen, Sapon Semsirmboon, Narongsak Chaiyabutr, Sumpun Thammacharoen

**Affiliations:** 1Department of Physiology, Faculty of Veterinary Science, Chulalongkorn University, Henri Dunang Street, Bangkok 10330, Thailand; 2The Academy of Science, The Royal Society of Thailand, Dusit, Bangkok 10300, Thailand; 3Queen Saovabha Memorial Institute, The Thai Red Cross Society, Bangkok 10330, Thailand

**Keywords:** DCAD, goat, heat stress, hypocapnia, urolithiasis

## Abstract

**Simple Summary:**

Urinary tract obstruction is a common problem in male goats. Urine acidification resulting from ammonium chloride treatment has been proved to dissolve sandy stones in alkaline urine. Although the natural high ambient temperature of the tropical area activates acid-base balance to alkalosis direction, a low dietary cation-anion difference regimen could acidify goat urine. The tubular excretion of acid and electrolytes plays a crucial role during mixed acid-base situations: respiratory alkalosis and metabolic acidosis.

**Abstract:**

Goats can suffer from intermittent heat stress in high ambient temperature (HTa) conditions, which causes sporadic respiratory hypocapnia. Obstructive urolithiasis is a common urological problem in goats. Sandy uroliths can be partially relieved by urine acidification with short-term ammonium chloride (NH_4_Cl) treatment. However, the outcome of urine acidification and the physiological responses to short-term NH_4_Cl supplementation under respiratory hypocapnia of HTa have rarely been reported. The present study investigated the effect of NH_4_Cl supplementation that produced a low dietary cation-anion different (l-DCAD) diet on acid-base balance and renal function under HTa conditions. The first experiment investigated the physiological responses to natural HTa to prove whether the peak HTa during the afternoon could induce HTa responses without a change in the plasma cortisol. The partial pressure of CO_2_ also tended to decrease during the afternoon. The second experiment examined the short-term effect of l-DCAD under HTa conditions. Although the blood pH was within the normal range, there was a clear acid-base response in the direction of metabolic acidosis. The major responses in renal function were an increase in tubular function and acid excretion. With a comparable level of creatinine clearance, the fractional excretions (FE) of chloride and calcium increased, and the FE of potassium decreased. Acid excretion increased significantly in the l-DCAD group. We conclude that under HTa conditions, the tubular excretion of electrolytes and acids was the major response to acid loading without changing the filtration rate. The l-DCAD formulation can be used to acidify urine effectively.

## 1. Introduction

Raising goats under high ambient temperature (HTa) conditions requires tailored management. Dairy goats and cows fed in tropical conditions at an HTa can suffer from heat stress [1,2], and, therefore, they require tailored management. According to Saipin et al. [1], heat stress is the second and third phase of HTa response that can be demonstrated mainly by the activation of the hypothalamic-pituitary-adrenal axis. This ambient condition is an environmental factor that can induce respiratory hypocapnia in dairy cows and goats [3,4]. This condition is the internal factor that determines the responses of acid-base homeostasis [4]. Sporadic respiratory hypocapnia, the decrease in blood partial pressure of CO_2_ due mainly to breathing, in livestock fed in tropical conditions is a fundamental factor in any management that influences acid-base homeostasis [1,2,3,4].

Urolithiasis is a common disease of the goat urinary tract. Male goats are at a high risk of obstructive urolithiasis [5,6]. The prognosis and surgical treatment outcomes of obstructive urolithiasis are poor [7,8,9]. The use of ammonium chloride (NH_4_Cl) to acidify urine and resolve sandy uroliths is usually the therapeutic recommendation before obstruction occurs [5]. In addition, the concept of dietary management to prevent stone formation has been proposed and partially investigated [10,11,12]. Adjustment of dietary cation and anion difference (DCAD) in feed has a potential benefit in ruminants. Under HTa conditions in tropical areas, high-DCAD formulations (39.0 mmEq/100 g DM) have been proposed to enhance heat dissipation and feed intake in dairy goats when compared with normal DCAD formulation (22.8 mmEq/100 g DM) [13,14]. Ammonium chloride treatment or supplementation of feed to achieve a low degree of DCAD (l-DCAD) formulation is well known for its capacity to acidify urine and has been recommended for the medical control of urolithiasis in goats [11,15]. However, under HTa conditions, the outcome of the l-DCAD feeding regimen and the response mechanisms can be affected by sporadic respiratory hypocapnia. Specifically, the sporadic respiratory hypocapnia that shifts the acid-base balance to the alkalosis stage may influence the metabolic acidosis induced by the l-DCAD regimen. This study aimed to investigate whether the l-DCAD regimen for goats with respiratory acid-base disturbance has the potential to acidify urine with the physiological response mechanisms of acid-base homeostasis.

## 2. Materials and Methods

### 2.1. Experimental Design and Animal Care

This study was performed at the Center of Learning Network for the Region of Chulalongkorn University in Nan Province. The animal use protocol was approved by the Institutional Animal Care and Use Committee of Chulalongkorn University Laboratory Animal Center (protocol number: 2131011). The procedures of this study were in compliance with the guidelines for the use of animals from the National Research Council of Thailand and followed the regulations and Animals for Scientific Purposes Act (A.D. 2015).

Twelve lactating Saanen goats aged 3 years with an average body weight of 43.9 ± 0.9 kg were enrolled. All goats were at their second lactation without health problems based on clinical signs and physical examination. The sample size was estimated according to the difference in physiological responses to HTa effect from our previous research [13]. Goats were individually housed in metabolic cages (2 × 1 m shaped cage with a plastic floor). The cages were in a barn with a natural light cycle. Concentrate was supplied twice daily at 07:00 and 14:00. Roughage (pangola hay) and water were freely accessible in the cage. Milking was performed twice daily at 06:00 and 13:00.

The experiment was conducted over 14 days during the mid-lactating period in which milk yield is at the plateau level, including a pretreatment period and a treatment period. During the pretreatment period, the goats (*n* = 12) received a control diet (normal DCAD) as a concentrate for seven days. The ambient conditions and the physiological parameters were collected to demonstrate the HTa effect. During the treatment period, goats were randomly allocated into control and treatment groups. One group (*n* = 6) received the control diet, and the other group received the l-DCAD diet. A low DCAD diet was formulated by adding 20 g of NH_4_Cl to 1 kg of the control diet. This intervention was based on the daily dose of NH_4_Cl that can decrease urine pH [15]. Both groups were fed their respective experimental diets for seven days. At the end of the experiment, each diet was collected and analyzed according to the Association of Official Analytical Chemists (AOAC) [16]. The chemical compositions of the diets are presented in Table 1.

### 2.2. Ambient Condition, Behavior, Sample Collections, and Measurements

Ambient temperature (Ta) and relative humidity (RH) were recorded daily at 06:00, 12:00, and 18:00 throughout the experimental period using a wet- and dry-bulb thermometer. The temperature and humidity index (THI) were calculated using Formula (1):THI = ((1.8 × Ta) + 32) − [(0.55 − (0.0055 × RH)) × (1.8 × Ta) − 26.8)](1)
where Ta: the ambient temperature (°C); RH: relative humidity (%).

Food and water were provided manually to each goat via separated buckets. Food intake and water intake (WI) were recorded daily. Dry matter intake (DMI) and WI were calculated from the amounts of feed (dry matter basis) and water based on the offer and refusal weights. Throughout the experiment, rectal temperature (Tr) and respiratory rate (RR) were measured two times daily at 06:00 and 18:00. On day seven, in both the pretreatment and treatment periods, blood samples were collected at 08:00 and 15:00 for blood gas, hematocrit, and plasma electrolyte analysis. Urine samples for kidney function evaluation were collected for 8 h from 09:00 to 17:00. Blood samples were collected from the jugular vein and divided into two parts. As is standard procedure for the blood gas sampling technique, the first part (1 mL) of blood was used for blood gas, glucose, and urea nitrogen measurements using an EG8 + iSTAT cartridge (Abbott, Freehold, NJ, USA). The hematocrit was measured from this sample after the blood gas measurements were carried out. The second blood sample (4 mL) was placed in a lithium-heparin tube and immediately placed into an icebox. The plasma was separated and aliquoted into four 1.5 mL tubes and stored at −20 °C for further plasma electrolyte, creatinine, and cortisol measurements. The concentrations of plasma sodium (Na) and potassium (K) were measured using a flame photometer (Flame photometer 410C, Ciba Corning Inc., Phoenix, AZ, USA). The plasma chloride (Cl) concentration was measured using a chlorimeter (chloride 925, Ciba Corning Inc., Phoenix, AZ, USA). The ionized calcium (Ca) concentration was measured using an automated analyzer (The IL ILab 650 Chemistry Analyzer, Diamond Diagnostic, Holliston, MA, USA). Creatinine concentration was measured using an automated analyzer (Mindray BS-800, Shenzhen Mindray Bio-Medical Electronics Co., Ltd., Shenzhen, China). The concentration of plasma cortisol was measured using the enzyme-linked immunosorbent assay technique (CBS-E18048G, CUSABIO, Houston, TX, USA) with intra- and inter-assay confidence interval values of less than 8% and 10%, respectively. The plasma osmolarity (Formula (2)) and anion gap (Formula (3)) were calculated as described by Stevens et al. (1994) [17].

osmolarity = 1.86 × [Na^+^ + K^+^] + [Glucose/18] + [BUN/2.8](2)

AG = [Na^+^ + K^+^] − [HCO_3_^−^ + Cl^−^](3)

To study kidney function, fresh urine samples were collected for 8 h using mineral oil-filled urine bags that could maintain anaerobic conditions. The total volume of urine was measured, and urine pH was immediately measured using a pH meter (Digicon PH-218, Sangchai Meter Co., Ltd., Bangkok, Thailand), and the rest of the urine (30 mL) was kept at −20 °C for analysis of the urinary electrolytes and creatinine using a procedure similar to that described above. The endogenous creatinine clearance and fractional excretion (FE) of the electrolytes were calculated using Formulas (4) and (5), respectively.
(4)$$\mathrm{Creatinine}\;\mathrm{clearance}=\frac{\left[\mathrm{Cr}\right]\mathrm{urine}*V}{\left[\mathrm{Cr}\right]\mathrm{plasma}}$$
where [Cr] is the concentration of creatinine in urine and plasma (mmol/L); V: the volume of urine within 8 h collection (mL/min).
(5)$$\mathrm{FE}\;\mathrm{of}\;\mathrm{electrolytes}\;=100\;\times \;\left(\frac{\left[X\right]\mathrm{Urine}}{\left[X\right]\mathrm{plasma}}\;\times \frac{\left[\mathrm{Cr}\right]\mathrm{plasma}}{\left[\mathrm{Cr}\right]\mathrm{urine}}\right)$$
where FE: the fractional excretion of electrolyte (%); [X]: concentration of electrolytes in the urine and plasma (mmol/L); [Cr]: concentration of creatinine in the urine and plasma (mmol/L).

Another part of the urine sample was determined for net acid excretion [18]. A mixture containing a urine sample (1 mL) and 0.1 N HCl (1 mL) was heated in a boiling water bath for 2 min to expel CO_2_. The mixture was then cooled for 10 min at room temperature (25 °C). The titration was conducted with 0.1 N NaOH to pH 7.4 at 37 °C. A blank (distilled water) was treated using a similar procedure. The titratable acid concentration (TA) was calculated based on the difference between the NaOH volume required to titrate the sample and the blank, as shown in Formula (6).
TA (mmol/L) = (∆ V_NaOH_) × 0.1_(N NaOH mol/L)_ × 1000(6)
where TA: the titratable acid concentration; ∆ V_NaOH_: the difference between the volume of NaOH used to titrate the sample and the blank.

Subsequently, 8% formaldehyde (5 mL) was added using the formal titration method to release H^+^ from the ammonium molecule (NH_4_^+^). This procedure would have reduced the solution pH, and the titration was continuously conducted until the pH was back to 7.4 using 0.1 N NaOH. The net acid concentration (NAC) was calculated from the final volume of sodium hydroxide used for titration from both the urine and blank using Formula (6). The concentration of ammonium was calculated from the difference between NAC and TA.

### 2.3. Statistical Analysis

The data are presented as mean ± SEM. The effects of either times or diets on ambient condition and physiological parameters and urinary electrolytes were compared using a *t*-test. Data that contain diet and time as the factors were analyzed with a repeated two-way analysis of variance. To control family-wise error, the significance of main effects was performed by the Bonferroni posttest. All statistical analyses were performed using Prism 5.0 (GraphPad Software, Inc., San Diego, CA, USA). Significance was declared at *p* < 0.05.

## 3. Results

During the pretreatment period, the ambient conditions, behavioral and physiological responses, plasma chemistries, electrolytes and hormones, and blood gas parameters were recorded in the morning and afternoon (Table 2). 

The amount of DMI, WI, and UV during this period was 1.42 ± 0.08, 4.74 ± 0.41, and 0.90 ± 0.19 kg/day, respectively. Morning ambient conditions, including Ta and RH (25 °C and 89%), were significantly different from the afternoon (31 °C and 74%). During the pretreatment period, the difference in Ta (ΔTa) between the afternoon and morning was approximately 6 °C. The afternoon THI (83.63 ± 0.89) was significantly higher than the morning THI (76.14 ± 0.59). The daily HTa effects on behavioral and physiological responses showed a significant increase in RR and Tr. Blood glucose from the afternoon sample was also significantly higher than that from the morning sample. Although there was a tendency toward a decrease in PCO_2_ (*p* = 0.057), the effect of the afternoon HTa condition on blood gas homeostasis was not significant for any of the parameters. Kidney function, including endogenous creatinine clearance, electrolyte FE, and NAC, was studied on day seven of the pretreatment period. The calculated endogenous creatinine clearance rate was 119 ± 12 mL/min. The FE values of Na, K, Cl, and Ca were 0.72 ± 0.10, 43.94 ± 3.15, 0.18 ± 0.03, and 0.73 ± 0.07%, respectively. The urine pH and total acid excretion values were 8.21 ± 0.03 and −178.6 ± 23.8 mmol/L. The net acid, titratable acid, and ammonium concentrations were −178.17 ± 23.79, 178.58 ± 23.84, and 0.42 ± 0.23 mmol/L, respectively.

The ambient conditions during the treatment period were similar to the pretreatment period; the Ta, RH, and THI from the morning and afternoon were 25.56 ± 0.18 °C, 91.64 ± 0.48%, and 77.13 ± 0.31 and 31.25 ± 0.73 °C, 69.88 ± 2.22%, and 83.26 ± 0.76, respectively (Figure 1a). l-DCAD had no effect on DMI, WI, or UV during the treatment period. The values of total daily DMI, WI, and UV from the control group were 1.34 ± 0.12 kg/day, 4.68 ± 0.50 L/day, and 1.00 ± 0.16 L/day, respectively. The values of these parameters from the l-DCAD group were 1.49 ± 0.11 kg/day, 6.96 ± 1.00 L/day, and 2.27 ± 0.68 L/day, respectively. Although there was no significant total daily WI or UV, an analysis of daytime and nighttime WI and UV that was conducted separately revealed that daytime WI in the control group (4.25 ± 0.45 L) was significantly lower (*p* < 0.05) than that in the l-DCAD group (6.50 ± 0.86 L) and nighttime UV in the control group (0.54 ± 0.10 L) was significantly lower (*p* < 0.05) than that in the l-DCAD group (1.36 ± 0.33 L). There was a prominent effect of HTa without the effect of l-DCAD on the behavioral and physiological responses, including Tr and RR (Figure 1b–c).

During the seven days of l-DCAD treatment, the plasma Cl concentration was significantly higher (Table 3). Low DCAD also influenced the blood pH, PCO_2_, HCO_3_, and base excess (Table 3). Moreover, low DCAD influenced kidney function by modifying tubular function without changing the filtration rate (Table 4). Endogenous creatinine clearance in the l-DCAD group was not different from that in the control group. The FE of K was lower; however, the FEs of Cl and Ca were higher in the l-DCAD group. Importantly, low DCAD decreased the urine pH significantly when the measurement was performed on day seven after treatment. The urine concentrations of Cl and Ca were higher, and the urine concentration of Na^+^ was lower in the l-DCAD group (Figure 2).

## 4. Discussion

The present study revealed the tendency for respiratory hypocapnia in dairy goats fed in HTa conditions in tropical areas and that l-DCAD supplementation in feed could acidify urine under respiratory hypocapnic conditions. The acid-base balance of the intravascular fluid and kidney function are the primary systems that respond to HTa and l-DCAD feeding. Within seven days of the l-DCAD regimen, the acid-base status shifted to the direction of acidosis, and the major kidney response was the activation of tubular function.

We first demonstrated the effect of short-term natural HTa in dairy goats by comparing morning and afternoon Ta. The gradual increase in Ta from early morning to afternoon in the present experiment produced ΔTa at approximately 5 °C. This value was lower than the natural ΔTa report at 7–10 °C [1,2,3,4,13]. The present study revealed a clear effect of HTa on respiratory rate and rectal temperature. The increase in both parameters was in line with a higher degree of ΔTa [1,4,13]. Unlike short-term HTa exposure in laboratory rats [19], Hct was maintained at a comparable value when Ta was increased during the afternoon. Our results suggest that HTa increased the core temperature, activated evaporative heat dissipation, and apparently maintained the intravascular fluid volume. Although blood gas parameters during the pretreatment period were maintained within the normal range [17,20], there was a tendency for hypocapnia due to the breathing effect in the afternoon HTa. Importantly, the present degree of ΔTa did not increase the plasma cortisol levels, suggesting that the hypothalamic-pituitary-adrenal axis was not activated [18]. Under the natural increase in Ta during the daytime, the ΔTa at 5 °C based on the baseline Ta at 25 ± 0.3 °C could activate physiological heat dissipation mechanisms without stress. Moreover, the condition shifted the acid-base status to the respiratory hypocapnia direction.

Supplementation of feed with ammonium chloride lowered the value of DCAD to less than zero in the present investigation. Under this l-DCAD feeding regimen for seven days, blood gas parameters were maintained within the normal range [17,20]. However, there were significant differences in all these parameters compared to the control feeding regimen (Table 3). The significantly lower values of pH, PCO_2_, HCO_3_, and Tot CO_2_ in the l-DCAD group suggested that the present l-DCAD feeding regimen shifted the systemic acid-base balance toward the metabolic acidosis direction. Interestingly, under HTa conditions during the treatment period, all animals were in the hypocapnia stage. The condition was prominent owing to both respiratory and l-DCAD-induced hypocapnia.

Another concurrent mechanism that is activated with an alteration in acid-base homeostasis is an increase in tubular function in the kidney. The l-DCAD regimen did not influence the plasma concentration of electrolytes (Table 3), except for in the plasma Cl concentration. In addition, the urine volume and endogenous creatinine clearance did not differ from those of the control group (Table 4). However, low DCAD significantly changed the FE of electrolytes, and the results were almost in line with previous experiments in which ammonium chloride was fed to goats directly [11,15]. The increased FE of Cl was apparently due to the higher plasma concentration of Cl and the balance between Cl and bicarbonate ions via the HCO_3_^−^/Cl^−^ exchanger, the chloride shift phenomenon. The reciprocal increase in plasma Cl ions and decrease in plasma bicarbonate ions were supported by similar anion gap levels. The FE of Na in the present experiment tended to be lower in the l-DCAD group, although the value was not significantly different from that of the control group. This result was in contrast to a previous report that direct ammonium chloride treatment increased the FE of Na in goats [11]. This was apparently due to the difference in the degree of ammonium-chloride-induced acidosis and the complex mechanisms of renal handling of Na during acidosis [21,22,23]. Our finding that the effect of the FE of Ca was increased under the l-DCAD feeding regimen was similar to previous reports [10,11,15,21,22] and suggests that under HTa conditions, Ca homeostasis is a major consideration in goats under an l-DCAD feeding regimen. The effect of ammonium-chloride supplementation on the FE of K^+^ was inconclusive [11,15,22]. The present finding of lower FE of K was probably due to a decrease in distal tubular secretion [22]. In addition to the effect of l-DCAD on the tubular function of electrolyte control, another straightforward tubular function during acid loading is an increase in acid excretion. However, the present study did not demonstrate an increase in ammonium excretion during l-DCAD-induced aciduria. This is in contrast to the usual findings of the previous reports on cattle [21,24,25]. We performed a positive control for the analytical procedure of acid excretion with human urine, and the proportion of ammonium excretion from human urine was within the normal range (Appendix A). Because most of the ammonia present in urine at pH 6.5–7.6 was an ammonium ion (NH^4+^), part of the hydrogen ion that is excreted in urine should be in this form [26]. Our interpretations of this discrepancy are that the degree of urinary pH (Table 4, pH = 6.82) in the current study was maintained at the inflection point of the sigmoid curve between pH and ammonium excretion [24], that there should have been an unidentified ammonium property in the goat urine, or perhaps that the major mechanism for acid excretion is not ammonium excretion from the kidneys of goats; however, this discussion is beyond the scope of the current study. Collectively, the prominent l-DCAD feeding regimen on kidney function could be demonstrated by tubular functions, including the excretion of electrolytes and acids. In addition, the l-DCAD effect, which did not influence urine volume and creatinine clearance, was partly explained by the maintenance of the filtration load.

## 5. Conclusions

The results of the present study show that the l-DCAD feeding regimen for seven days is an effective procedure for acidified urine under HTa-induced respiratory hypocapnia. The major responses of the kidneys were increased acid excretion and electrolyte handling. Calcium excretion and homeostasis should be the main considerations when implementing this feeding regimen for goat farming.

## Figures and Tables

**Figure 1 animals-12-03444-f001:**
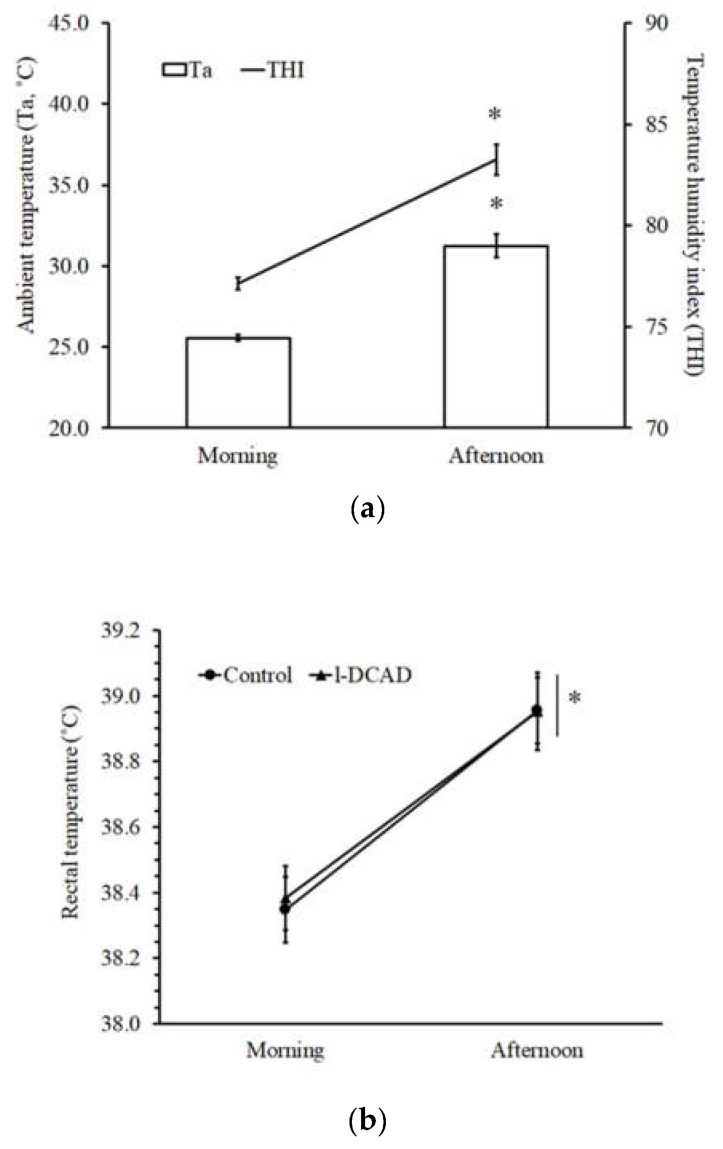
The effect of high ambient temperature on the biological parameters. (**a**) The natural ambient condition at the experimental area between early morning (0600) and afternoon (1800), including ambient temperature (Ta) and the temperature and humidity index (THI). (**b**) Rectal temperature and respiratory rate (**c**) were increased significantly; * the significant effect of time.

**Figure 2 animals-12-03444-f002:**
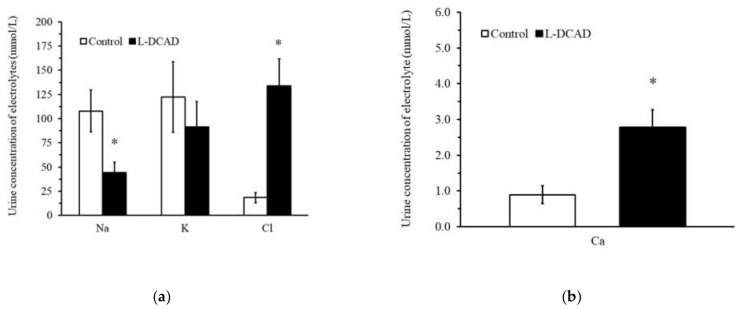
The effect of l-DCAD on (**a**) urinary concentration of sodium, potassium, and chloride and (**b**) calcium; * the significant effect of l-DCAD.

**Table 1 animals-12-03444-t001:** Feed chemical compositions of experimental diets (dry matter basis).

Chemical Compositions, % DM	Roughage	Control Feed	L-DCAD Feed
Moisture	7.32	8.74	8.70
Dry matter	92.68	91.26	91.30
Crude protein	3.87	14.67	17.44
Fat	0.95	3.57	3.83
Ash	7.15	6.50	5.51
Calcium	0.73	1.05	0.92
Phosphate	0.12	0.50	0.37
Sodium	0.32	0.03	0.05
Potassium	1.42	1.28	1.28
Sulfur	0.19	0.19	0.11
Chloride	0.30	0.54	1.43
Neutral detergent fiber	73.12		
Acid detergent fiber	44.84		
DCAD	30.00	7.04	−12.16

DCAD in milliequivalents of (Na + K) − (S + Cl)/100 g of DM.

**Table 2 animals-12-03444-t002:** Ambient parameters and biological parameters of dairy goats measured in the morning and afternoon during the pretreatment period.

	Morning	Afternoon	*p*-Value
Ambient condition			
Ta	25.13 ± 0.3	31.00 ± 0.67	<0.01
RH	89.25 ± 2.77	74.31 ± 2.98	<0.01
THI	76.14 ± 0.59	83.63 ± 0.89	<0.01
Biological parameters			
Respiratory rate (bpm)	30 ± 3	53 ± 4	<0.01
Rectal temperature (°C)	38.5 ± 0.1	39.5 ± 0.1	<0.01
Hct (%)	22.3 ± 0.6	22.0 ± 0.6	0.60
Glucose (mg/dL)	55.6 ± 0.8	58.8 ± 1.1	0.05
BUN (mg/dL)	19.3 ± 1.0	18.6 ± 1.2	0.54
Na (mmol/L)	148.5 ± 1.4	150.3 ± 1.4	0.38
K (mmol/L)	3.76 ± 0.10	3.82 ± 0.14	0.72
Cl (mmol/L)	103.9 ± 0.7	104.8 ± 0.5	0.30
Ca (mmol/L)	1.89 ± 0.05	1.95 ± 0.05	0.26
Cortisol (ng/mL)	42.76 ± 9.61	37.13 ± 9.18	0.33
pH	7.43 ± 0.01	7.44 ± 0.01	0.28
PCO_2_ (mmHg)	40.47 ± 0.57	38.76 ± 0.74	0.057
HCO_3_ (mmol/L)	26.80 ± 0.68	26.28 ± 0.53	0.34
Tot CO_2_ (mmol/L)	27.92 ± 0.68	27.33 ± 0.53	0.27
Base excess (mmol/L)	2.33 ± 0.83	2.17 ± 0.56	0.80
Anion gap (mmol/L)	21.5 ± 1.7	23.1 ± 1.5	0.54

Blood for analysis was collected at 08:00 and 15:00. Other data were collected at 06:00 and 18:00. Ta = ambient temperature (°C), RH = relative humidity (%), and THI = temperature humidity index. The HTa effect is the effect of time, where the afternoon Ta was significantly higher than the morning Ta.

**Table 3 animals-12-03444-t003:** The effects of l-DCAD and HTa on the biological responses of dairy goats.

	Morning		Afternoon		SEM	Effect (*p* Value)	
	Control	l-DCAD	Control	l-DCAD		Time	l-DCAD
Blood chemistries, electrolytes, and hormones							
Glucose (mg/dL)	53.5	53.2	60.8	58.7	7.94	<0.01	0.42
BUN (mg/dL)	20.8	19.0	20.7	20.8	1.98	0.18	0.77
Na (mmol/L)	150.9	149.3	150.1	153.5	18.12	0.32	0.72
K (mmol/L)	3.72	4.08	3.66	4.04	0.13	0.74	0.12
Cl (mmol/L)	102.0	107.8	101.2	109.8	4.54	0.52	<0.01
Ca (mmol/L)	1.76	1.75	1.93	1.89	0.01	<0.01	0.64
Cortisol (ng/mL)	56.91	86.30	64.92	52.56	5100	0.77	0.66
Blood gas parameters							
pH	7.42	7.38	7.46	7.38	0.0004	0.03	0.02
PCO_2_ (mmHg)	40.58	37.93	39.12	37.02	1.24	0.03	0.02
HCO_3_ (mmol/L)	26.57	22.27	27.70	22.10	2.05	0.43	<0.01
Tot CO_2_ (mmol/L)	27.83	23.33	28.83	23.33	1.90	0.40	<0.01
Base excess (mmol/L)	2.17	−3.17	3.67	−2.83	2.54	0.19	<0.01
Anion gap (mmol/L)	26.0	23.3	24.9	25.6	17.87	0.74	0.76

Blood for analysis was collected at 08:00 and 15:00.

**Table 4 animals-12-03444-t004:** The effects of l-DCAD on the kidney functions of dairy goats.

	Control	l-DCAD	*p*-Value
Endogenous creatinine clearance			
Plasma creatinine (mg/dL)	0.60 ± 0.05	0.53 ± 0.06	0.40
Urine creatinine (mg/dL)	50.88 ± 15.47	53.07 ± 19.57	0.93
Urine volume (8 h, mL/min)	1.65 ± 0.31	2.85 ± 1.01	0.28
Creatinine clearance (mL/min)	113.03 ± 23.95	159.40 ± 20.29	0.17
Fractional excretion of electrolytes, %			
Na	1.05 ± 0.05	0.42 ± 0.42	0.053
K	40.91 ± 3.55	23.31 ± 3.17	<0.01
Cl	0.21 ± 0.04	1.38 ± 0.14	<0.01
Ca	0.89 ± 0.25	2.78 ± 0.45	<0.01
Acid concentration (mmol/L)			
Net acid	−159.97 ± 28.53	−5.33 ± 14.87	<0.01
Titratable acid	−159.97 ± 28.53	−18.00 ± 8.94	<0.01
Ammonium	0.00 ±0.00	12.67 ± 7.56	0.12
Urine pH	8.21 ± 0.09	6.82 ± 0.44	0.01

## Data Availability

The data used to support the findings of this study are available from the corresponding author (ST) upon request.

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
