# Peer review of "Effects of Low Dietary Cation and Anion Difference on Blood Gas, Renal Electrolyte, and Acid Excretions in Goats in Tropical Conditions"

_animals, 2022, doi:10.3390/ani12233444_

Round 1

Reviewer 1 Report

Dear authors,

Τhe present text presents many omissions and inadequate methodologies.

Firstly, there is no definition of heat stress in the introduction of your research. How sure are you that your animals (which obviously are well adapted) are subjected to heat stress in the conditions described?

Sample size calculation and eligible criteria are considered decisive in the design of a clinical trial.

Supplementation of a different diet in animals for further investigation should always applied with a adjustment period. The difference that you recorded may derived by just from the change of ratio.

Several other limitations are highlighted in the attached paper 

Author Response

------------------------------------

Reviewer #01 (23 comments)

Comment01: These two sentences need to be combined!

Ans: Yes, it’s done: The first experiment investigated the physiological response of natural HTa to prove whether the peak HTa during the afternoon could induce HTa responses without a change in the plasma cortisol.

Comment02: the sentence ".....with the normal range of ...." has no meaning

Ans: We agree with reviewer and the phrase has been deleted.

Comment03: dairy goats and cow suffer from heat stress and therefore they requite tailored management.

Ans: Thank you very much for this point. We added “and therefore they requite tailored management” in the sentence.

Comment04: the term hypocapnia should be explained.

Ans: Yes, it’s done: Sporadic respiratory hypocapnia, the decreased in blood partial pressure of CO2 due mainly to breathing, in livestock fed in tropical conditions….[1-4].

Comment05: need reference justification + poor English

Ans: We do add 4 references from our group to support this argument. The problem of “poor English” will be send to the recommend “language editor center” later after the manuscript has been accepted by the journal. Thank you very much.

Comment06: need to be explained (what does high refer to? quantity? high cation or high anion?

Ans: We add the information of DCAD value from our experiments. These include both high DCAD and normal DCAD. That is “Under HTa conditions in tropical areas, high-DCAD formulations (39.0 mmEq/100 g DM) have been proposed to enhance heat dissipation and feed intake in dairy goats when compared with normal DCAD formulation (22.8 mmEq/100g DM) [13-14].”

Comment07: Need justification, explanation and reference

Ans: This is our rationale. The idea is straightforward that if the balance of acid-base is influence by HTa, then the outcome and mechanism should be changed or different from the animal living in the temperate zone. We then hypothesized whether the l-DCAD formulation could change urine pH in goat fed under natural HTa condition and we focused on the physiological responses and acid-base balance. We appreciate and agree with reviewer suggestion that this should be explain in more detail. The additional information has been done. That is “However, under HTa conditions, the outcome of the l-DCAD feeding regimen and the response mechanisms would be affected by sporadic respiratory hypocapnia. Specifically, the sporadic respiratory hypocapnia that shift the acid-base balance to the alkalosis stage may influence the metabolic acidosis induced by l-DCAD regimen.”.

Comment08: Eligible criteria age, primiparous, mulitparous, breed, previous medical history....etc  should be reported. The sample size need justification

Ans: The information has been added: All goats were at the second lactation without health problems based on clinical signs and physical examination. The sample size was estimated according to the difference in physiological responses of HTa effect from our previous information [13].

Comment09: period characteristics?

Ans: The information has been added: The experiment was conducted over 14 days during mid-lactating period that milk yield is at the plateau level, including a pre-treatment period and a treatment period.

Comment10: How the allocation was conducted?

Ans: The information has been added: During the treatment period, goats were randomly allocated into control and treatment groups.

Comment11: In what was this intervention based on?

Ans: Thank you very much for this good question. We have added more information that describe this intervention. That is “A low DCAD diet was formulated by adding 20 g of NH4Cl to 1 kg of the control diet. This intervention based on the daily dose of NH4Cl that could decreased urine pH [15].”.

Comment12: the abbreviation should be explained

Ans: Done: Association of Official Analytical Chemists

Comment13: Poor English

Ans: Although we have sent our manuscript to the “Elsevier language editing center” before submit to “Animals” as the attached file (certificate), we do accept this weakness. However, we try to revise this. In addition, we agree to re-send this manuscript to the recommend “language editor center” later after the manuscript has been accepted by the journal. Thank you very much.

Comment14: Tr definition?

Ans: We do apologize for this mistake. The full term has been included.

Comment15: RR definition

Ans: We do apologize for this mistake. The full term has been included.

Comment16: Poor English

Ans: Although we have sent our manuscript to the “Elsevier language editing center” before submit to “Animals” as the attached file (certificate), we do accept this weakness. However, we try to revise this as much as we can. In addition, we agree to re-send this manuscript to the recommend “language editor center” later after the manuscript has been accepted by the journal. Thank you very much.

Comment17: A brief description should be added

Ans: Done: The plasma osmolarity (osmolarity = 1.86 x [Na+ + K+] + [Glucose/18] + [BUN/2.8]) and anion gap (AG = [Na+ + K+] – [HCO3- + Cl-]) were calculated as described by Stevens et al. (1994) [17].

 In addition, the reference number has been re-assigned here. This is our mistake, sorry.

Comment18: should be deleted

Ans: Done.

Comment19: Statistical analysis report has to be enriched in what format the data were collected ? which Statistical package was used?  Why bonferroni test was applied?

Ans: The information has been added: The data are presented as the mean ± SEM. The effects of either times or diets on ambient condition and physiological parameters (table 2) and urinary electrolytes (figure 2) were compared using T-test. Data that contain diets and time as the factors was analyzed with the repeated two-way analysis of variance. To control family-wise error, the significance of main effects was performed by the Bonferroni posttest. All statistical analysis was done using Prism 5.0 (GraphPad Software, Inc., San Diego, CA, USA). Significance was declared at P < 0.05.

Comment20: This should be reported in material and methods

Ans: This is true, thanks. The information “The ambient conditions and the physiological parameters were collected to demon-strate the HTa effect” has been added in M&M section.

Comment21: group?

Ans: Yes, done.

Comment22: is The last column l-DCAD the p value?

Ans: Yes, we have modified the table and added this information.

Comment23: this table should be presented in the results.

Ans: Yes, the data was there at the last row “urine pH”. However, the curve of ammonium chloride excretion was from ref [24].

Reviewer#01: Comments and Suggestions for Authors

Firstly, there is no definition of heat stress in the introduction of your research. How sure are you that your animals (which obviously are well adapted) are subjected to heat stress in the conditions described?

Ans: We have added the definition of heat stress in the revised version based on our previous results that is “According to saipin et al [1], heat stress is the second and third phase of HTa responses that mainly can be demonstrated by the activation of hypothalamic-pituitary-adrenal axis.”. In addition, in the present manuscript we do not need to have “heat stress” condition. We do demonstrate that “respiratory hypocapnia” which probably could interfere the feeding regimen.

Sample size calculation and eligible criteria are considered decisive in the design of a clinical trial.

Ans: We have provided this information in the revised version: All goats were at the second lactation without health problems based on clinical signs and physical examination. The sample size was estimated according to the difference in physiological responses of HTa effect from our previous information [13].

Supplementation of a different diet in animals for further investigation should always applied with a adjustment period. The difference that you recorded may derived by just from the change of ratio.

Ans: Thank you very much for this critical suggestion. This experiment aimed at investigate the short term responses of blood gas and kidney functions. In addition, all animals were on the control diet normally. 

Reviewer 2 Report

I have indicated all comments, suggestions for revision in the manuscript (animals-2004546-peer-review-v1.pdf).

Author Response

--------------------------------------

Reviewer #02 (35 comments)

Comment01&2: Strikethrough Text & The manuscript describes a study on lactating goats so it would be appropriate to describe how high temperature and acid-base imbalance affect the physiological state of goats during this period.

Ans: This is true. We actually introduce some information on the earlier paragraph. We agree to put more detail in revised version. The sentence “This condition would be the internal factor that determine the responses of acid-base homeostasis [4]” has been added. In addition, the present manuscript is about the effect of l-DCAD supplementation during HTa. In fact, l-DCAD has been used as the nutritional management of urolithiasis in goat. Then, we think that the information of “urolithiasis” in also important. To compromise this point, we have revised this information and hope that this should be OK. That is “Urolithiasis is a common disease of the goat urinary tract. Male goats have a high risk of obstructive urolithiasis [5-6]. The prognosis and surgical treatment outcomes of obstructive urolithiasis are poor [7-9].”. Thank you very much for this valuable suggestion.

Comment03: Please add the age of the goats.

Ans: Yes, it’s done: Twelve lactating Saanen goats aged 3 years with an average body…..

Comment04: Please describe where the cages were? Were they in one room, or just under a roof, or in an open space, etc.? What was the light cycle.

Ans: The information has been added. “The cages were within the barn with natural light cycle.”

Comment05:  Feed ....

Ans: Yes, thank.

Comment06: dry matter basis

Ans: Yes, thank.

Comment07: Were there automatic drinkers and feeders in the cages?

Ans: The detail information of food and water was added; Food and water were provided to each goat via separated bucket manually.

Comment08: Was urine collected throughout the treatment period or on the seventh day of the period?

Ans: The 24 h urine volume (UV) from each goat was collected throughout experiment. We used this data to verify our 8 h UV that we studied kidney functions. Although we get the significant correlation between 24 h and 8 h UVs and actually the information could go to “supplement material”, we find some weak point for 24 h UV collection technique. This weakness came from the difficulty of 24 h UV collection per se. After discussed with our team, we would like to withdraw this sentence from the manuscript. Thank you very.

Comment09: Please write the full name of these abbreviations.

Tr -  rectal temperature

RR - respiratory rate

Ans: Yes, done.

Comment10: What time was the third measurement? O 12:00?

Ans: Actaully, it’s 2 times.

Comment11-12: Strikethrough Text & The results of the milk parameters are not discussed in this manuscript.

Ans: Yes, we agree to delete this.

Comment13: Ambient parameters and biological parameters of dairy goats measured in the morning and afternoon during the pre-treatment period.

Ans: Thank you very much for this suggestion. We do change in revised version.

Comment14: Specification

Ans: Yes, the information was added: Ta = ambient temperature (°C), RH = relative humidity (%), THI = temperature humidity index.

Comment15-18: Biological parameters, Biological parameters (including blood) & Strikethrough Text

Ans: Yes. We agree with this and make change as suggestions.

Comment19-20: Blood for analysis was collected at 8:00 and 15:00.  Other data were collected at 0600 and 1800.

Ans: Yes, thank you very much.

Comment21: A legend is absent.

Ans: Yes, we do change the new picture.

Comment22-23: biological parameters

Ans: Yes, done

Comment24-25: Strikethrough Text & Blood plasma parameters

Ans: Actually, we would like to preserve this detail because we would like to emphasize between “blood biochemistry, electrolytes and hormone” and “blood gas parameters”.

Comment26-27: Blood parameters  

Ans: Actually, we would like to preserve this detail because we would like to emphasize between “blood biochemistry, electrolytes and hormone” and “blood gas parameters”.

Comment28-29: Blood for analysis was collected at 0800 and 15:00. & Highlight text

Ans: Yes, we agree with this suggestion.

Comment30: A legend is absent.

Ans: Yes, we do change the new picture.

Comment31-32: Strikethrough Text & respiratory rate and rectal temperature.

Ans: Yes, thanks.

Comment33-34: Strikethrough Text & in both parameters

Ans: Yes, thanks.

Comment35: This publication was probably included by mistake.

Ans: This is our mistake. We agree to delete the reference.

Round 2

Reviewer 1 Report

Dear Authors,

The revised version is presented improved. All my recommendations were fulfilled. 

In line 3 of introduction saipin et al should be written with "S".

Nice work

Author Response

Dear Reviewer 1

    Thank you very for your help. according to your suggestion belew, I have corrected this in the revised version.

Suggestion: In line 3 of introduction saipin et al should be written with "S"

Ans: Done with yellow hightlight, thanks 

Sincerely yours

Sumpun